# Ruminations Regarding Characteristics of Quintessential Adult Communicative Play

**DOI:** 10.3390/bs16010002

**Published:** 2025-12-19

**Authors:** John O. Greene

**Affiliations:** Brian Lamb School of Communication, Purdue University, West Lafayette, IN 47907, USA; jgreene@purdue.edu

**Keywords:** adult communicative play, assembly difficulties, bullying, conversational enjoyment, “ideal” play, theory of transcendent interactions, social skill

## Abstract

Greene and Pruim’s (2023) theory of adult communicative play (TACP) was developed as an effort to address considerations of: (a) pattern and novelty, (b) interpersonal connection, and (c) enjoyment as they pertain to adult conversational activities by recourse to a parsimonious, integrated conceptual framework. Central to their treatment is the notion of quintessential (or “ideal”) play, referring to occasions characterized by: (a) receptivity and absorption in the conversation; (b) comprehension and understanding; (c) connection and mutuality; and (d) a sense of discovery and insight. This conception of “ideal play” is viewed as the endpoint of a continuum along which efforts at play may be understood to be successively less and less mutually enjoyable as one moves away from the “ideal” endpoint. The primary aim here is to further refine and clarify the nature of quintessential play. In particular, “ideal” play is posited to: (1) unfold over multiple, mutual conversational entries, (2) be relatively rare and fleeting, (3) be enacted in pursuit of the enjoyment derived from the communicative event itself, (4) involve mutual, improvisational contributions to the interaction, and (5) be both the product and source of enhanced communication skills.

## 1. Introduction

Adults *do* play. Indeed, adult play is exceedingly common and manifests in myriad forms, among them, communicative play – instances of which seeemingly present themselves at every turn. Further, not only is communicative play common, it is held to serve a number of individually- and socially-based functions (see, for example: [1]; [4]; [7]; [9]; [12]; [13]; [16]; [17]; [25]; [27]; [37]; [44]; [45]; [49]; [52]). Despite its ubiquitious nature, and salutary effects, adult communicative play remains a relatively understudied (and, even more, under-theorized) area of inquiry. The aim of this essay is to refine and extend a particular theoreitcal appproach to the topic and to highlight certain methodological implications of that treatment for future research.

In tracing the contours of their conception of “adult communicative play”, [24] ([24]) give emphasis to three aspects of that phenomenon. They note that their focus is on: (a) interpersonal processes (i.e., interactive rather than solitary activities), (b) that are constituted by verbal and nonverbal message features, and (c) that are enacted in pursuit of the pleasures inherent in the communicative event itself (i.e., “play for its own sake”). The upshot of this construal of adult communicative play is to center attention on *conversational activities* (with lesser concern for other sorts of playful endeavors such as athletic competition, card games, chess, etc.).[note 1] Examples, then, of activities that *do* fall within the domain of adult communicative play are numerous and varied, including sharing jokes and puns, playful teasing, originating sexual invitations and euphemisms, collaborative storytelling and embellishment, and use of imitative vocalic and facial cues, among many others.

## 2. Properties Commonly Ascribed to Adult Communicative Play

Survey of the literature on adult play, communicative and otherwise, suggests various recurrent themes, among them: (a) pattern and novelty, (b) interpersonal connection, and (c) enjoyment (see [24]). Regarding the first of these, it is not uncommon to find conceptual treatments that make explicit the idea that instances of play very often involve capitalizing on the routine in creative ways (e.g., [5]; [9]; [17]; [25]; [34]). With respect to notions of interpersonal connection, there is also recurrent recognition of the role of interdependence, rapport, and so on, in fostering occasions of play (e.g., [3], [4]; [6]; [7]; [49]). Finally, the most frequent observation regarding the nature of adult play is almost certainly that such occasions are marked by pleasure and enjoyment (see, for example, [3]; [10]; [15]; [17]; [25]; [30]; [35]; [38]; [39]; [40]; [41]; [49]).

## 3. The Theory of Adult Communicative Play

The prominence of conceptions of pattern and novelty, interpersonal connection, and enjoyment in treatments of adult play served as impetus for the development of the theory of adult communicative play ([24])—an effort to address these characteristics of playful interactions by recourse to a parsimonious, integrated conceptual framework. The theory of adult communicative play (TACP), itself, draws upon the theory of transcendent interactions (TTI; [21]; see also [23]), a conceptual formulation—centered on experiences of interpersonal engagement that constitute “the very extreme of human symbolically based connection and receptivity” (Greene & Herbers, p. 81)—that gives emphasis to “conjoint mentation” or the “socially interactive nature of thinking” (Greene & Herbers, p. 67).

TTI ([21]) specifies four properties of the most extreme cases of collaboratively driven, interpersonal engagement. “Receptivity and absorption” refers to conditions of maximal conversational attentiveness and immersion. “Comprehension and understanding” involves the twin notions that, on one hand, a person is able to fully grasp another’s meaning, and on the other, that his or her interlocutor understands and appreciates one’s own conversational entries. “Connection, mutuality, and sharedness” concerns “mutual participation in a coordinated interleaving of conversational and conceptual contributions” (Greene, p. 49). Finally, “exploration, discovery, and insight” refers to experiences of novelty and fresh perspective that arise in the course of an interaction.

Turning to TACP, the aforementioned themes of pattern and novelty, interpersonal connection, and enjoyment that run through the literature on adult play have readily apparent parallels with the fundamental tenets of TTI—and of particular relevance, the TTI conceptions of *assembly difficulties* and the proposition that *the resolution of assembly difficulties is inherently pleasurable* (see [24], pp. 300–301). In essence, interactions that people find to be the most engaging and evocative are those in which conversational partners both pose and assist in resolving, assembly difficulties—a dynamic that is held to be inherently hedonically positive.[note 2]

According to the TACP framework, the best occasions of communicative play (i.e., “ideal play”) are defined as “those central exemplars of code-based (verbal and nonverbal) instances of interpersonal engagement, understanding, and coordination, marked by a sense of novelty and enjoyment, that are understood to count as ‘play’ by the participants themselves” ([24], p. 302, emphasis deleted). TACP treats “adult communicative play” as a “fuzzy set”—a class of phenomena without clearly defined boundaries, but with more central exemplars, gradually shading to cases that are progressively less and less characteristic of that category ([46]; [51]). The upshot of this line of thinking is the notion of a continuum (actually a “ray”—a line with a clearly defined endpoint that extends infinitely in one direction), bounded at one end by instances of quintessential (i.e., “most perfect,” “pure”) or “ideal play,” and experiences that are less and less play-like as one moves away from that endpoint.

The major components of TACP, then, concern the nature of the factors that determine how close instances of communicative play come to the endpoint of the “play continuum”—i.e., what conditions function to foster exchanges that more nearly approximate quintessential or “ideal play.” On this point, TACP distinguishes “contextual” features—background conditions that facilitate playful interactions, and “processual” features—characteristics of interactions that make them more or less “play-like.” Contextual features, then, include: (a) person factors (e.g., personality traits, message-production and processing skills), (b) relationship factors (e.g., affection, power), (c) dyadic factors (e.g., shared experiences, similarity in preferences for certain types of play), and (d) environmental and situational factors (e.g., social norms). Turning to processual factors (again, characteristics of conversational exchange that enhance their “playful quality”), TACP centers on the roles of: (a) mutuality (e.g., mutual enjoyment, mutual behavioral enactment versus “one-sided” efforts at play), and (b) improvisation (i.e., novelty, “freshness” versus repetition and routinization).

## 4. The Nature and Research Implications of “Ideal Play” and the “Play Continuum”

### 4.1. Ideal Play—Sustained Exchanges

The conception of adult communicative play as a fuzzy set poses the question of the nature of the features that typify the central exemplars of that category. On this point, it is instructive to consider the characteristics of occasions that are most likely to foster experiences of: (a) receptivity and absorption; (b) comprehension and understanding; (c) connection and mutuality; and (d) exploration, discovery, and insight specified in TTI. I would posit, then, that the “best examples” of playful occasions are those that are carried out over multiple conversational turn-changes. In contrast to a single humorous conversational entry (e.g., use of a familiar playful nickname, allusion to some past shared funny event or joke), experiences of shared immersion and novelty are more likely to arise when one’s actions are picked up and extended by his or her another interlocutor.

Again, in contrast to instances of communicative play extending over multiple, mutual conversational entries, a conversational entry may occur simply in passing, understood and appreciated by the other, but lacking the deep sense of engagement and freshness that characterizes “ideal play.” Similarly, even a pervasive atmosphere of relational playfulness (perhaps arising from shared dispositional playfulness, see [24]) may lead to frequent occasions of conversational quips and fun, but again, viewed through the TTI/TACP lens, it should be apparent that “ideal play” involves deeper, sustained engagement by the parties involved.[note 3] For researchers, the cautionary note here is to keep in mind that many instances of playful acts may fall into ranges of the “play continuum” that do not fully reflect the quintessential endpoint of that continuum.

### 4.2. Ideal Play—Rare and Short-Lived

A second implication of grounding the conception of adult communicative play in the TTI/TACP framework is to suggest that experiences that mark the very endpoint of the play continuum may be both rare and fleeting. In part, this point derives from the Action Assembly Theory (AAT2; [18], [19], [20]) characterization of the evanescent nature of the thought processes that underlie overt verbal and nonverbal behavior; from [19] ([19], pp. 144–145):

One moment I have in mind what I intend to do; the next, I’m thinking of the mole on the other person’s cheek; and then again of my plan for talk. I may at some point have in mind a representation of what I’m doing or saying, but that particular action specification may not persist for the duration of its execution…. I’m suggesting a characterization of messages and message-encoding processes in which mental states and entities are seen as evanescent, fast, shifting, and parallel, where overt message components may be disjointed and incoherent, where actions are specified at multiple representational levels, and where the mechanisms that govern the interplay of thoughts and actions are seen as essential concerns.

A second line of theorizing extends this conception of constantly evolving malleable message-relevant cognitions to explicitly address the transitory nature of conversational engagement. From the perspective of TTI ([21]; [23]), the fast, fluid nature of the message-production system suggests that maximal interaction engagement (i.e., “receptivity and absorption”) is unlikely to be sustained over extended periods. That is, at one moment, the contents of conscious awareness may be fully focused on the unfolding interaction, but other processes will inexorably draw an individual’s thoughts away from full cognitive engagement in the unfolding conversation (see [19]). Put simply, periods of immersive interaction may be fleeting, and even when play continues to carry and characterize the interaction, other processes will draw a person’s thoughts in other directions.

In essence, receptivity and absorption is a matter of degree (think of the fluid nature of the contents of consciousness[note 4]), and seen from this perspective, the temporal characteristics of playful interactions, and the position along the play continuum of particular moments that make up those interactions, become necessary components of theorizing about the phenomenon (i.e., what factors will contribute to driving attentional focus toward or away from the unfolding interaction?). Beyond the role of individual-difference variables in sustaining intense occasions of conversational engagement (see [23]), the answers to this question will require consideration of situational exigencies, personal goals and motivational factors, and, almost certainly, the content and sequencing of the message exchange itself.

### 4.3. Ideal Play—Play for the Sake of Play

At the outset of their theoretical formulation [24] ([24]) make explicit that their point of conceptual departure is interpersonal exchanges “whose proximal focus and purpose is the communicative event itself—that is, play for its own sake” (p. 298)—a frequent theme in the literature on play (e.g., [17]; [35], [36]; [38]; [50]). As [36] ([36], p. 42), for example, puts it, “[B]eing in a playful mode means forgoing any definite aim or pursuit.”

It is important to note, however, that (1) play may arise in the pursuit of other conversational aims and activities, and (2) playful interchanges may afford satisfaction associated with other intra- and interpersonal functions as well. Thus, beyond the positive hedonic tone of playful interactions driven by the dynamics of assembly processes, it may also be true that there are occasions where play gives rise to enjoyment because it serves to facilitate self-expression, foster closer interpersonal relationships, and so on.[note 5]

The idea that pleasure associated with play-like activities may stem from distinct underlying processes is in keeping with the commonly recognized fact that the brain is an evolutionary kluge ([31]; [32])—a hodgepodge of ad hoc systems that have developed over the course of human evolution. Rather than a seamless, integrated system, the brain consists of relatively more recent anatomical components overlaying ancient structures (see [14]; [28]). It should not be surprising, then, that different systems (and the, perhaps clunky, functional interplay of those systems) may play a role in the subjective hedonic experiences associated with play.

And herein lurks an issue that theorists and researchers need to keep in mind. Given two alternative theoretical formulations, one predicated on the thesis that resolution of assembly difficulties is inherently pleasurable, and another emphasizing goal accomplishment/need fulfillment (along with attendant processes of planning, formulation of outcome expectations, etc.), an obvious course of action would be to develop a contrastive test of these alternative accounts. But the difficulty here is that to demonstrate that one set of processes is in play does not necessarily rule out the functioning of alternative systems.[note 6]

### 4.4. Ideal Play—Mutual Improvisation

A key element of [24]’s ([24]) theorizing is to give emphasis to the: (a) mutual and (b) improvisational nature of “ideal play.” “Mutuality” here can be seen to derive from the “comprehension and understanding” and “connection, mutuality, and sharedness” properties ascribed to the best instances of communicative play. When people perceive that they fully grasp their partner’s meaning, and that the other is similarly able to follow and appreciate their own message cues, and when there is a sense of conjoint participation in the unfolding interaction, instances of play are likely to prove more satisfying. Put simply, occasions of quintessential play are marked by mutual participation, understanding, and enjoyment, and as play becomes increasingly one-sided, the interaction will fall farther and farther from the endpoint of the play continuum.[note 7]

The “improvisational” character of quintessential conversational play is related to experiences of “exploration, discovery, and insight” identified in TTI ([21]).[note 8] Certainly, although there may be enjoyments associated with enactments of repetitive routines and references, such conversational moves may shade become into the mundane and boring, and the “best” instances of play capitalize on the “old” in fresh new ways.[note 9]

And, there is an even larger point to be made here concerning mutuality and improvisation. We should recognize that adult communicative play may be characterized by mutuality and improvisation—leading to pleasure and enjoyment—and still fall short of the qualities that mark the ultimate endpoint of the play continuum. “Ideal play” is not simply improvisation; rather, it is improvisation marked by a sense of discovery, insight, and new understanding. Furthermore, “ideal play” is not simply overtly manifested and observable mutuality of conversational entries and responses; rather it further involves a sense of sharedness—a deep experience of connection with another human being. These points suggest that investigations of adult communicative play need to reflect a cognizance of the *subjective qualities* pertaining to mutuality and improvisation inherent in quintessential play.

### 4.5. Ideal Play—A Product of Skill/A Producer of Skill

As noted at the outset, scholars commonly recognize that adult communicative play serves a number of important individually and socially-based functions. Moreover, as with virtually any domain of communicative endeavor (whether it be public speaking, listening, assertiveness, detection of deception, or what have you), people differ in their capacity to successfully engage in playful conversation (see [6]; [11]; [33]; [35]). On one hand, then, we can think of ideal communicative play as the product of people’s skill repertoire. On balance, those with more extensive declarative and procedural memory resources ([2]; [47]; [48]) are more likely to hit upon ways of finding pleasure in their conversations with others.

It is also the case that conversational play sets the stage for expanding one’s skill repertoire (see [6]; [9]; [17]; [25]). In the course of play, people are able to “try out” new identities and to explore previously untapped avenues of interpretation and action—a dynamic that is likely enhanced in those instances of play where there is little perception of threat ([35]). Moreover, the propensity for acquisition of new skills in the course of play should be particularly pronounced when new behavioral options satisfy recurrent assembly difficulties. [21]’s ([21]) “principle of resolution-based acquisition” posits that people are especially receptive to “solutions” to assembly problems that they have either hit upon themselves or observed on the part of others. As he notes (p. 47), “[W]e tend to notice, retain, and subsequently implement and re-implement (and refine, adapt, modify) ways of handling recurrent exigencies.”

## 5. Methodological Considerations

An essential criterion in evaluating any theoretical formulation centers on heurism, and it is presumably apparent that the research implications of the TACP are numerous. From Section 3 above, any number of hypotheses concerning the role of “contextual features” (i.e., person factors, relationship factors, dyadic factors, and environmental/situational factors) and “processual features” (i.e., mutuality and improvisation) spring readily to mind. Of course, testing TACP-derived hypotheses necessitates operationalizing the adult play construct. And there do exist measures of that phenomenon (see [6]; [8]; [42]).[note 10] In our own research, we are currently employing two approaches to operationalization: (1) a six-item Likert-type self-report measure (e.g., “We both felt a sense of humor and laughter,” “I experienced a real sense of play and fun”), and (2) narrative reports of people’s experiences of “occasions of most enjoyable communicative play” and “occasions of ‘failed’ communicative play.”

One of the aims of this essay, though, is to advance a refined conception of adult communicative play that carries with it operational implications. Future research efforts might profitably take into account that instances of play can be arrayed along a continuum, from quintessential experiences (i.e., characterized by “absorption,” “understanding,” “mutuality,” and “discovery”) to those that are progressively less ideal. And, going forward, we should keep in mind that the quintessential endpoint of the play continuum is most likely to be reflected in rare and short-lived (rather than sustained) exchanges enacted for the sake of play itself, arising from mutual improvisation.

## 6. Conclusions

It stands to reason that a great deal of the theorizing and research on interpersonal communication has focused on the often-problematic aspects of human interaction. After all, conflict, social anxiety, bullying, and so on seem to demand attention (presumably in the hope that things can be improved). Consequently, there is something of a taken-for-granted quality for those aspects of interpersonal communication that proceed relatively smoothly. Nevertheless, there is increasing awareness that, despite their lack of the “slap-in-the-face” quality of communication-gone-bad, there are important things—processes impinging on the quality of our relationships and our personal well-being—that merit our attention.

The current treatment reflects an effort to identify the characteristics of one of these “taken-for-granted” social processes: adult communicative play. In the same way that scholars might seek to identify those qualities that mark mastery or exceptional performance in other realms of communicative endeavor, there may be something to be learned from pursuing analogous issues as they apply to the domain of communicative play. But, in all honesty, just as defining exemplary performance in other communicative domains establishes a benchmark for describing and illuminating less-than-optimal occasions, a clear depiction of the qualities that characterize quintessential conversational play establishes a conceptual foundation for understanding instances where things don’t go so well.

## 7. Epilogue

A reviewer gently persuaded me that I wasn’t quite finished, so let me end with a few additional ruminations. One should come at any theory with a certain degree of skepticism—and this is particularly true of the theorist him or herself. In the present case, there are various theoretical claims about which I am not totally convinced, including even the most central proposition: that the resolution of assembly difficulties is intrinsically gratifying. A key implication of this point centers on its universality—it should apply to everyone, everywhere. But it is the case that much of the research on the topic of play and humor has focused on differences in forms and preferences for types of play. We should note, though, that there is no inherent contradiction at hand; where one person might find enjoyment in the resolution of assembly difficulties associated with teasing and playful banter, another might delight in creative sarcasm.

And here we begin to paw at a deeper issue centering on the sources of individual differences in play manifestations and preferences. In pursuit of this issue, the literature is replete with studies involving the “usual suspects”—personality traits (e.g., the “Big 5,” “humor styles,” “dispositional playfulness”) and “culture” (e.g., “individualist” vs. “collectivist”) (and throw in sex/gender, if you like). But the problem with recourse to explanations based on traits and culture is that they afford a rather limited account of the generative processes involved. As [26] ([26], pp. 150, 152) observed regarding trait-based explanations:

To say that leaves in northern climates have a predisposition to fall from trees in autumn is not a very satisfying explanation. To say that a person avoids communication because they have a disposition to do so is equally unrewarding…. Whenever trait theorists have been pressed to look for the mechanisms *behind* that trait, mechanisms outside the realm of predispositions that almost always involve processes of interpretation and production of behavior…

And a similar argument applies to appeals to cultural factors. At some point, addressing individual differences in communicative play will require examination of the cognitive processes underlying message production and interpretation. Despite the presence of a particular personality trait, a person may or may not exhibit behavioral manifestations in line with that predisposition. Despite being fully enculturated in a given worldview, a person’s communicative behavior in any particular instance may not reflect the norms of his or her reference group(s). What factors, then, are in play in situations such as these? And, I would assert that the same mechanisms are in play when people *do* behave in line with their traits and cultural modes of thinking and acting. As [29] ([29], p. 341) has noted, it is not particularly problematic if an explanation leaves something *else* to be explained, but “An explanation leaves something to be desired when there is something *more* to be explained—details to be filled in, a closer approximation arrived at…” Scholars interested in adult communicative play might well avail themselves of models of memory structures and processes, skill acquisition, and the like.

Finally, I would end with the admonition to keep in mind (from Section 4.2) that the notion of “quintessential adult communicative play” is, most probably, exceedingly rare—a “conceptual ideal” that may not often occur (or, if it does, it might not last for very long). But as an “ideal type”, it serves to anchor our understanding of an entire class of communication phenomena.

## Data Availability

No new data were created or analyzed in this study. Data sharing is not applicable to this article.

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
