# Peer review of "Ruminations Regarding Characteristics of Quintessential Adult Communicative Play"

_behavsci, 2025, doi:10.3390/bs16010002_

Round 1

Reviewer 1 Report

Comments and Suggestions for Authors

We understand that the present manuscript is based on a long experience the author has with the topic and with the TACP and ideal play. 

However, it is not clear to us what it is the purpose of this manuscript. It is no research question, no specified aim, no introduction. The term Introduction is missing, looking like not complte article (with the missing parts in the first part). 

The references are rather old, not being clear how the article relates to actual state of debate, except the own elaboration of the author: only 2 titles are from the last five years, one of them being the article of the author. 

Please make more explicit what is the purpose of this article, what resaerch gap it addresses, and what is the purpose of the theoretical considerations listed.  

Reviewer 2 Report

Comments and Suggestions for Authors

I greatly appreciate the contribution to defining a new construct and the theory of adult communicative play (TACP), which could be associated with the perspective of positive psychology.

I  do not agree with the way th author use the word quintessential, wich does not mean ideal, and cannot be related with the word ideal, in my oppininon. It means the essence, the main thing, the essential in a conception, in a doctrine, in a work... In fact, in Greek philosophy, it meant the material principle of the world, attribute of the ether, considered by ancient cosmogony as the fifth element (in addition to earth, water, air and fire). Than mean that it is very important! So way ”ideal”?

The study is the result of a literature review, but for now, the validity of this approach is insufficient and will need to be empirically verified. How can the construct be measured and how can the theory be empirically tested? 

Also, I needed to read a short sections about the relation of this new construct with other (apparently) close concepts : adult play, adult playfulness etc., and with different associated notions, such as satisfaction with life, humour,  mental health etc. 

Wich are the main functions of this construct in the interpesonal relationship (friends, colleagues) in  comparaisson with romantic / marital relationships?

Round 2

Reviewer 1 Report

Comments and Suggestions for Authors

The revisions bring more clarifications to the essay. The further clarifications to the elaborated model are welcomed. However, more considerations about the strengths of the model and his empirically grounded elaboration should be emphasised. Also, some cautions, or limits of the model, should be highlighted. 

Author Response

The revisions bring more clarifications to the essay. The further clarifications to the elaborated model are welcomed. However, more considerations about the strengths of the model and his empirically grounded elaboration should be emphasised. Also, some cautions, or limits of the model, should be highlighted. 

Thanks you for your work in reviewing this paper (twice!). In this revision I have added a new section, "7. Epilogue," in which I address some limitations and reservations about the approach and raise an additional point or two about directions for future research and theory in the area.